# Pyrrolizine/Indolizine-NSAID Hybrids: Design, Synthesis, Biological Evaluation, and Molecular Docking Studies

**DOI:** 10.3390/molecules26216582

**Published:** 2021-10-30

**Authors:** Mohammed A. S. Abourehab, Alaa M. Alqahtani, Faisal A. Almalki, Dana M. Zaher, Ashraf N. Abdalla, Ahmed M. Gouda, Eman A. M. Beshr

**Affiliations:** 1Department of Pharmaceutics, Faculty of Pharmacy, Umm Al-Qura University, Makkah 21955, Saudi Arabia; 2Department of Pharmaceutical Chemistry, Faculty of Pharmacy, Umm Al-Qura University, Makkah 21955, Saudi Arabia; amqahtani@uqu.edu.sa (A.M.A.); famalki@uqu.edu.sa (F.A.A.); 3Sharjah Institute for Medical Research, University of Sharjah, Sharjah 27272, United Arab Emirates; U17105878@sharjah.ac.ae; 4Department of Pharmacology and Toxicology, Faculty of Pharmacy, Umm Al-Qura University, Makkah 21955, Saudi Arabia; anabdrabo@uqu.edu.sa; 5Department of Pharmacology, Medicinal and Aromatic Plants Research Institute, National Center for Research, Khartoum 2404, Sudan; 6Medicinal Chemistry Department, Faculty of Pharmacy, Beni-Suef University, Beni-Suef 62514, Egypt; 7Department of Medicinal Chemistry, Faculty of Pharmacy, Minia University, Minia 61519, Egypt; eman_beshr@mu.edu.eg

**Keywords:** pyrrolizine, indolizine, NSAIDs, cytotoxicity, cell cycle, apoptosis, docking study

## Abstract

In the current study, eight new hybrids of the NSAIDs, ibuprofen and ketoprofen with five pyrrolizine/indolizine derivatives were designed and synthesized. The chemical structures of these hybrids were confirmed by spectral and elemental analyses. The antiproliferative activities of these hybrids (5 μM) was investigated against MCF-7, A549, and HT-29 cancer cell lines using the cell viability assay, MTT assay. The results revealed 4–71% inhibition of the growth of the three cancer cell lines, where **8a**,**e**,**f** were the most active. In addition, an investigation of the antiproliferative activity of **8a**,**e**,**f** against MCF-7 cells revealed IC_50_ values of 7.61, 1.07, and 3.16 μM, respectively. Cell cycle analysis of MCF-7 cells treated with the three hybrids at 5 μM revealed a pro-apoptotic increase in cells at preG1 and cell cycle arrest at the G1 and S phases. In addition, the three hybrids induced early apoptotic events in MCF-7 cells. The results of the molecular docking of the three hybrids into COX-1/2 revealed higher binding free energies than their parent compounds **5a**,**c** and the co-crystallized ligands, ibuprofen and SC-558. The results also indicated higher binding free energies toward COX-2 over COX-1. Moreover, analysis of the binding modes of **8a**,**e**,**f** into COX-2 revealed partial superposition with the co-crystallized ligand, SC-558 with the formation of essential hydrogen bonds, electrostatic, or hydrophobic interactions with the key amino acid His90 and Arg513. The new hybrids also showed drug-likeness scores in the range of 1.06–2.03 compared to ibuprofen (0.65) and ketoprofen (0.57). These results above indicated that compounds **8a**,**e**,**f** deserve additional investigation as potential anticancer candidates.

## 1. Introduction

Currently, more than 150 anticancer drugs approved by the FDA for the treatment of different types of cancer are available in the market [1]. However, the rates of morbidity and mortality due to cancers are still high, even within high-income countries [2]. In addition, the low rate of survival in patients with metastatic cancers [3] also confirms the urgent need for continuous research in this field.

Treatment of cancer with a single drug was associated in many cases with the development of resistance and low therapeutic outcomes [3,4]. To overcome this problem, several approaches have been developed in cancer therapy. Among these approaches, combination therapy was emerged as a promising approach in cancer therapy [5]. The combined drugs can be used to target different essential pathways in cellular proliferation, where they can act either synergistically or additively to combat the overgrowth of cancer [6,7].

Combination therapy has showed preferential advantages over the single-agent treatment for survival rates and the tumor response [8]. Due to the promising outcomes of combination therapies, a huge number of clinical trials were launched to investigate combination therapies in cancer and other diseases [9]. However, combination therapy also has several disadvantages. Among these, the higher side effects which develop after administration of two or more drugs compared to the single-agent treatment [10]. Combination therapy also produced higher toxicity than the single agent treatment [8]. Pharmacokinetic problems may also appear on administration of combined drugs since management of the pharmacokinetics of a single drug is easier. In addition, the potential for drug-drug interactions in combination therapy is higher than the single-agent treatment [11].

Hybrid/conjugate drugs also emerged as a promising strategy in cancer therapy [12,13]. They could provide a solution to the poor patient acceptability to combination therapy [13,14]. The hybrid/conjugate drugs could also provide several advantages over the traditional combination therapy including the lower toxicity and side effects [15,16]. In addition, the management of the pharmacokinetics of a single agent could be easier.

NSAIDs are also considered one of the cancer pain medications [17]. They constitute a key component of the WHO analgesic leader to control cancer pain [18]. However, besides their use as analgesic anti-inflammatory agents, NSAIDs have also displayed a chemopreventive effect in colorectal and breast cancer [19,20].

Our literature review revealed several acetylsalicylic acid-metal complexes with potent anticancer activity. The Se-aspirin complex **I** exhibited potent anticancer activity against a panel of 6 cancer cell lines with IC_50_ values in the range of 1.3–4.4 μM [21], Figure 1. Mechanistic studies of **I** into colorectal cancer cells revealed cell cycle arrest at G1 and G2/M phases and induction of apoptosis. In addition, Weninger et al. [22] have reported a series of acetylsalicylic acid derivatives with Zeise’s Salt (**IIa–d**). These derivatives exhibited growth-inhibitory activities against HT-29 and MCF-7 cancer cells at IC_50_ values in the range of 30-50 μM. In addition, Baecker et al. [23] have reported a series of cobalt alkyne complexes with fluorinated acetylsalicylic acid **III**. These complexes showed cytotoxic and apoptotic activities against the COX-1/2-expressing cancer cell lines (HT-29 and MDA-MB-231). On the other hand, compounds **III** exhibited much lower activity against MCF-7 cells which lack COX protein.

In addition, several successful examples of NSAIDs hybrids/conjugates in cancer research have been also reported, Figure 1. Although the parent NSAIDs have showed weak cytotoxic activities [24,25,26,27], their hybrids with the cytotoxic agents were reported to have potent anticancer activity. Among these hybrids, the riboflavin-dexibuprofen conjugates **IVa–e** were synthesized by Banekovich et al. [24]. A biological evaluation of these conjugates **IVa–e** revealed growth-inhibitory activity of MCF-7 and HT-29 at IC_50_ values of 8–15 µM. These results indicated that the conjugates have much higher antiproliferative activities than their parent components, dexibuprofen and tetraacetylriboflavin.

Curci et al. [25] have synthesized prodrug conjugate **V** of kiteplatin Pt(IV) and two molecules of ibuprofen. The new conjugate exhibited much higher cytotoxic activity compared to the parent components. It inhibited the growth of HCT15 and HCT116 cancer cell lines at IC_50_ of 0.45 and 0.26 µM, respectively.

The ibuprofen-podophyllotoxin conjugate **VI** was reported among a small series of podophyllotoxin designed by Zhang et al. [26]. Evaluation of cytotoxicity of **VI** against Bel-7402 and Bel-7402/5-FU cancer cell lines revealed IC_50_ values of 18.88 and 10.28 µM, respectively. These results indicated higher cytotoxic potential toward the resistant cancer cell line.

The hybrid **VII** was designed by combining 5,16-pregnadiene and ibuprofen [27]. Evaluation of the antiproliferative activity of **VII** revealed higher growth-inhibitory activity against the human anaplastic astrocytoma (U373) cells than the parent compounds.

### Rational Design

Previously, we reported compound **VIIIa–c** (Figure 2) with in vivo anti-inflammatory activities comparable with that of ibuprofen [28,29]. Only two of these hybrids (**VIIIb**,**c**) were also investigated for their cytotoxic activity against three cancer cell lines (IC_50_ = 0.06–0.87 μM). These results motivated us to investigate the structure-activity relationship and the potential mechanism which mediates the cytotoxic activity of this scaffold.

Many of the NSAIDs displayed weak antiproliferative activity which could be COX-dependent or independent [30]. Esterification/amidation of the carboxylic acid group in ibuprofen was associated in some cases with increases in the anti-inflammatory activity and decrease in GIT side effects [31,32]. Some of the ester and amide derivatives of ibuprofen also showed improved antiproliferative activity compared to the parent drug [30]. On a cellular level, this improvement of the antiproliferative activity could be due to the increased lipophilicity and enhancer cellular uptake [33].

Encouraged by the above findings, we have designed a new series of hybrids (scaffold **B**) by incorporating the pyrrolizine/indolizine derivatives (scaffold **A**) with NSAIDs, Figure 3. Ibuprofen was selected as the NSAID component of the new hybrids based on the promising cytotoxic activity of **VIIIb**,**c**. In addition, ketoprofen was also selected in this study to compare the effect of the two drugs on the antiproliferative activity of the new hybrids. The aim of this selection was to compare the impact of the two drugs on cytotoxic activities of the final hybrids, Figure 3.

To investigate the relationship between the chemical structure and the antiproliferative activity of the new hybrids (scaffold **B**), several pyrrolizine/indolizine derivatives (scaffold **A**) were used. In addition, the new hybrids were designed bearing electron-donating (OCH_3_) and electron-withdrawing (F, Cl, and Br) substituents at the *para*-position of ring A to compare the impact of the electronic effects of their antiproliferative activity, Figure 3. The new hybrids will be evaluated for their antiproliferative activity against a panel of cancer cell lines. An additional investigation will also be performed to evaluate the potential mechanism of their action of the most active compounds.

## 2. Results and Discussion 

### 2.1. Chemistry

The synthesis of the target hybrids **8a–i** is presented in Figure 1. Compounds **2a**,**b** were obtained from the reaction of **1a**,**b** with malononitrile according to the previous reports [34,35]. In addition, compounds **4a–d** were prepared from the reaction of chloroacetyl chloride with the appropriate aniline derivatives **3a–d** [34,35]. On the other hand, the pyrrolizine/indolizine derivatives **5a–e** were prepared from the reaction of **2a**,**b** with the appropriate acetanilide derivatives **4a–d** [36].

To prepare 4-methoxy/fluoro indolizine derivatives, **2b** was also refluxed with the appropriate acetanilide **4a**,**b** for 48 h. However, the duplication of the reflux time was also associated with the formation of uncyclized derivatives rather than the target indolizines. The IR spectra of the obtained products revealed absorption bands of the geminal cyano groups, which confirms the formation of uncyclized products. These results align with the results of our previous report [35].

Preparation of the target hybrids **8a–i** was achieved in two steps following the previous report [29]. In the first step, the NSAIDs, ibuprofen and ketoprofen were converted into their acid chloride derivatives **7a**,**b** using thionyl chloride. The second step takes place through an acylation of the primary amino groups in compounds **5a–e** using the freshly prepared acid chloride derivatives **7a**,**b**. Structural elucidation of the new hybrids **8a–i** was performed. The spectral data of compounds **8a**,**b** is discussed below.

The IR spectrum of pyrrolizine-ibuprofen **8a** revealed absorption bands at 2221 and 1659 cm^−1^ indicate the cyano and carbonyl groups, respectively (Appendix A). The ^1^H-NMR spectrum of **8a** revealed seven signals at the aliphatic region (*δ* 0.88–4.29 ppm) which indicate the aliphatic protons, (Appendix A). The two methyl groups of the isobutyl moiety (-CH_2_CH(CH_3_)_2_) appears as a doublet signal at *δ* 0.88 ppm. Another doublet signal at *δ* 1.61 indicates the methyl group of the propionamide moiety (-COCHCH_3_). In addition, four doublet signals are found at *δ* 6.74, 7.04, 7.21, and 7.26 ppm indicate the two *para*-substituted phenyl rings. Two singlet signals at *δ* 8.21 and 9.03 ppm indicate the two amide protons.

On the other hand, the IR spectrum of the ketoprofen hybrid **8b** revealed absorption bands at 2211 and 1698 cm^−1^, indicating the cyano and carbonyl groups, respectively (Appendix A). A total of 6 signals at the aliphatic region (*δ* 1.62–4.27 ppm) in the ^1^H-NMR spectrum of **8b** are observed indicating the aliphatic protons, (Appendix A). The signal corresponding to the methyl group of the propionamide moiety (-NHCOCHCH_3_) appears as a doublet signal at *δ* 1.62 ppm. The spectrum also shows one triplet signal at *δ* 2.83, which indicates the methylene group (CH_2_-1) of the pyrrolizine nucleus. A multiplet at *δ* 6.69–7.68 ppm indicates 12 of the aromatic protons of the 3 phenyl rings, while a single signal at *δ* 7.86 ppm indicates the aromatic proton at *ortho*-position of the phenyl propionamide ring (B). In addition, 2 singlet signals at *δ* 8.94 and 9.02 ppm indicates the two amide protons in this compound.

Appendix A including all spectral data and copies of IR, ^1^H-NMR, and ^13^C-NMR spectra of the final compounds **8a–i** are provided with this manuscript (Appendix A).

### 2.2. Biological Evaluation

#### 2.2.1. Antiproliferative Activity

##### Screening Assay

Previously, the ibuprofen hybrids/conjugates **IV**–**VIII** exhibited antiproliferative activities against different types of cancer cell lines [24,25,26,27,28,29]. Accordingly, the new hybrids **8a–i** were evaluated for their antiproliferative activity using the cell viability assay, MTT assay following the previous report [37]. In this assay, the new hybrids were evaluated against three cancer cell lines including MCF-7 breast, A549 lung, and HT-29 colon cancer cell lines. The antiproliferative activities of the starting materials **5a–e**, ibuprofen and ketoprofen, were also investigated in this assay. The cancer cells were treated with the test compounds at a single dose (5 μM) for 72 h. The inhibition in the viability of the cancer cell lines was calculated relative to the untreated control. The results (growth%) are represented in Figure 4.

The results of the cell viability assay of the tested hybrids against MCF-7 cells revealed 12–71% inhibition of the cancer cells’ viability, Figure 4, which were much higher compared with those of ibuprofen (5%) and ketoprofen (9%). The changes in the cell viability of MCF-7 cells treated by all the new hybrids **8a–i** were statistically different from the control (*p* < 0.01). Compounds **8a**,**e**,**f** induced more than 50% inhibition of the growth of MCF-7 cells at 5 μM. In addition, the pyrrolizine derivatives **8a–f** showed higher growth-inhibitory activities than their parent compounds **5a–c**, however, the indolizine derivatives **8g–i** exhibited lower growth-inhibitory activities than their parent compounds **5c**,**d**. The results also indicated higher antiproliferative for the ibuprofen derivatives (**8a**,**c**,**g**) compared to their corresponding ketoprofen analogs (**8b**,**d**,**h**). Among the new hybrids, **8f** exhibited the highest antiproliferative activity (71%) against MCF-7 cells, which was statistically different from those of the control and ibuprofen (*p* < 0.01).

The effects of the new hybrids **8a–i** on the growth of A549 cancer cell line are presented in Figure 5. Except fo**r 8h**, the new hybrids exhibited 13–48% inhibition in the growth of A549 cells compared to the control (*p* < 0.01). The results also revealed higher antiproliferative activity for the ibuprofen hybrids than the ketoprofen analogs. However, no antiproliferative activity was observed for ibuprofen and ketoprofen against A549 cells. Among the new hybrids, **8a**,**e**,**f** showed the highest antiproliferative activity.

Finally, compounds **8a–i** were also investigated for their effects on the viability of HT-29 cancer cells. The results are presented in Figure 6. Except for compound **8b**, all of the tested hybrids included 4–53% inhibition in the viability of HT-29 cells. However, both ibuprofen and ketoprofen did not show antiproliferative activity against HT-29 cells at 5 μM.

The results also indicated higher antiproliferative activity for the ibuprofen derivatives (**8a**,**c**,e,**g**) against HT-29 cells than their ketoprofen analogs (**8b**,**d**,f,**h**). Among the new hybrids, **8a**,**e**,**f** exhibited the highest antiproliferative activity (47–49%) against HT-29 cells compared to the control (*p* < 0.01).

In conclusion, the results of the cell viability assay showed the highest sensitivity for MCF-7 cells toward the new hybrids. Moreover, compounds **8a**,**e**,**f** were the most active in inhibiting the proliferation of MCF-7 cells.

##### Cytotoxicity Assay

Based on the results of the MTT viability assay (Figure 4, Figure 5 and Figure 6), three of the new hybrids (**8a**,**e**,**f**) were selected for further investigation of their antiproliferative activities. The MTT assay was used to determine the IC_50_ values of the selected hybrids following the previous report [38]. MCF-7 cell line, the most sensitive to the antiproliferative effect of the new hybrids, was selected for this study. Doxorubicin was used as a reference drug. The cancer cells were treated with the test compounds at different concentrations. The results expressed as IC_50_ values were determined, Table 1.

The results of the MTT assay revealed that the three hybrids inhibit the growth of MCF-7 cells at IC_50_ values in the range of 1.07–7.61 μM compared to doxorubicin (IC_50_ = 2.07 μM). The IC_50_ value of doxorubicin was slightly higher than the reported value [39]. Among the three compounds, **8e** was the most active in inhibiting the growth of MCF-7 cells, while **8a** showed the lowest cytotoxicity.

#### 2.2.2. Cell Cycle Analysis

Previous reports indicated that the amide derivatives of ibuprofen/ketoprofen induced cell cycle arrest at the G0/G1 or G1/S phases in cancer cell lines [30].

In the current study, the effect of compounds **8a**,**e**,**f** on the cell cycle distributions of MCF-7 cells was investigated **8a**,**e**,**f** were also investigated. The cancer cells were treated with the test compounds at 5 μM for 48 h. This study was performed following the previous report [36]. The results are presented in Figure 7.

The result of cell cycle analysis revealed that each of the 3 compounds showed profound increase in the preG1 phase in MCF-7 cells, as an indication of their pro-apoptotic activities, especially **8e** that showed 10-fold increase compared to control, Table 2. Additionally, **8a** and **8e** arrested cells in the S phase at the expense of G1 and G2/M phases, while **8f** slightly increased cells in the G1 phase.

#### 2.2.3. Annexin V-FITC/PI Apoptosis Assay

The amide derivatives of ibuprofen/ketoprofen were also reported to induce apoptosis in different types of cancer cell lines [30]. In the current study, annexin V-FITC/PI double staining protocol was used to investigate the effects of compounds **8a**,**e**,**f** on apoptotic events in MCF-7 cells. The cancer cells were treated with each of the 3 hybrids at 5 μM for 48 h. The study was performed following the previous report [36]. The results were presented in Figure 8.

The 3 hybrids **8a**,**e**,**f** induced increase in the early apoptotic events in MCF-7 cells ranging from 18–50 folds (11–30%) compared to the control, Table 3. This was at the expense of late and necrotic events which were absent.

### 2.3. Computational Studies

#### 2.3.1. Molecular Docking Studies

The main mechanism of action that mediate the anti-inflammatory activity of ibuprofen/ketoprofen is the reversible inhibition of COX-1/2 isoenzymes [40,41]. In addition, the mechanistic studies of the starting material **5c** revealed moderate inhibition of COX-1/2 with weak selectivity toward COX-2 [28]. Although several reports indicated that the antiproliferative activity of NSAIDs are COX-independent, some other reports described a correlation between COXs inhibition and this activity [30]. Moreover, many of the NSAIDs prodrugs have also displayed intrinsic anti-inflammatory activities mediated even in part by COXs inhibition [42].

Based on the above mentioned finding, we performed a molecular docking study of compounds hybrids **8a**,**e**,**f** into the two COXs to compare the binding characteristics against those of their parent compounds (**5a**,**c** and ibuprofen). The docking study was performed into the crystal structure of COX-1 (pdb 1EQG) [43] and COX-2 (pdb 1CX2) [44] using AutoDock 4.2 [45]. The ligands (**8a**,**e**,**f**) and protein molecules were prepared into the suitable formate (pdbqt) for AutoDock following the previous report [46]. In the current study, COX-1/2 proteins were used as rigid molecules, while the ligands were docked as flexible molecules. The 2/3D binding modes of the tested compounds were generated by Discovery Studio Visualizer [47].

Firstly, ibuprofen was re-docked into the active site of COX-1 to validate the docking procedures. The 2/3D binding modes of the re-docked ibuprofen was investigated against those of the co-crystallized ligand. A RMSD of 0.87 Å was observed between the re-docked and the co-crystallized ligand. The re-docked molecule showed identical hydrogen bonds and hydrophobic interactions with those of the co-crystallized ibuprofen, Appendix A.

Validation of the docking study into COX-2 (pdb: 1CX2) was also performed. the co-crystallized ligand, SC-558 was docked into the active site of COX-2. The results revealed superposition of the re-docked SC-558 with the co-crystallized ligand with RMSD of 1.30 Å. The binding mode of the re-docked SC-558 overlaid with the co-crystallized ligand is shown in Appendix A.

The results of the docking study of the 3 hybrids (**8a**,**e**,**f**) into COX-1 binding free energies in the range of −10.36 to −10.55 kcal/mol compared to ibuprofen (energy (Δ*G_b_* = −8.43 kcal/mol), Table 4. On the other hand, they exhibited higher binding free energies toward COX-2 (Δ*G_b_* = −10.70 to −12.56 kcal/mol) compared to −10.78 kcal/mol for SC-558. The binding free energies of the three hybrids were also higher than their parent compounds **5a**,**c**.

Compound **8a** exhibited a binding free energy (Δ*G_b_* ) of -10.36 kcal toward COX-1, which was higher than ibuprofen, Table 4. An investigation of the binding mode of **8a** revealed that the best fitting conformation adopted a binding orientation superposing with the co-crystallized ibuprofen into COX-1, where the 4-methoxyaniline moiety in **8a** superposed with the 4-isobutylphenyl moiety in ibuprofen, Figure 9. In addition, the carboxamide moiety at C2 of compound **8a** was also superposed with the propionic acid moiety in ibuprofen. This orientation allowed **8a** to interact similarly to ibuprofen with Arg120 and Tyr355 in COX-1 forming a cluster of four conventional hydrogen bonds. Compound **8a** and ibuprofen exhibited similar hydrophobic interactions with Val116, Val349, Ala527, and Leu531.

The results of the docking study of compound **8a** into COX-2 also revealed a higher binding free energy than SC-558. These results indicated that the hybrid **8a** has higher affinity toward COX-2 over COX-1. An analysis of the binding mode of the best fitting conformation of compound **8a** into COX-2 revealed nice superposition of the two phenyl rings with the phenyl rings of SC-558, Figure 10. The pyrrole ring in **8a** was also located near the binding position of the pyrazole ring. Compound **8a** exhibited one conventional hydrogen bond with Arg120 and one carbon hydrogen bond with Ser353. In addition, **8a** showed one electrostatic interaction with Arg513. Based on these results, the higher binding free energy of **8a** toward COX-2 could be attributed to the electrostatic interaction with Arg513 and the multiple hydrophobic interactions with the hydrophobic residues in COX-2.

Compound **8e** also showed higher affinity toward COX-1 than ibuprofen. However, **8e** did not show any conventional hydrogen bonds with the amino acids in COX-1, Figure 11.

The results of the docking study of **8e** into COX-2 also revealed higher binding free energy compared to COX-1. The 4-bromophenyl moiety in **8e** superposed with the bromophenyl moiety in SC-558 into the hydrophobic pocket of COX-2, Figure 12. In addition, the pyrrolizine nucleus was also overlaid with the pyrazole ring in SC-558, while the phenyl propionamide in **8e** extended with the phenyl sulfonamide moiety in SC-558 into the side pocket of COX-2. Compound **8e** displayed six hydrogen bonds including four conventional hydrogen bonds with His90, Leu352, and Arg513 and two carbon hydrogen bonds with His90 and Gly526, Figure 12.

Among all the tested compounds, **8f** displayed the highest binding free energy toward COX-2, Table 4. An investigation of the binding interactions of **8f** revealed one conventional hydrogen bond with Arg120 and one carbon hydrogen bond with Val116, Figure 13.

The orientation of the best fitting conformation of **8f** into COX-2 also revealed partial overlay of the two phenyl rings with the two phenyl rings in SC-558, Figure 14. The bromophenyl moiety in **8f** occupied the hydrophobic pocket of COX-2 and formed identical hydrophobic interactions with Leu384, Tyr385, and Trp387. On the other hand, the superposition of the phenyl carboxamide at C2 in **8f** with the phenyl sulfonamide moiety in SC-558 into the side pocket of COX-2 allowed the two molecules to form one electrostatic interaction with His90 and two hydrophobic interactions with Ser353 and Val523. Moreover, **8f** showed one carbon hydrogen bond with Arg120 compared to unfavorable donor-donor interaction for SC-558.

The 2/3D binding modes of the parent compounds (**5a**,**c**) were illustrated in Appendix A.

In conclusion, the three hybrids **8a**,**e**,**f** exhibited higher free energies toward COX-1/2 than the parent compounds (**5a**,**c**) and the co-crystallized ligands (ibuprofen and SC-558). In addition, their binding free energies toward COX-2 were higher than those for COX-1. An analysis of the binding modes of these hybrids into COX-2 revealed partial superposition with SC-558 and formation of hydrogen bonds, electrostatic, or hydrophobic interactions with the key amino acids His90, and Arg513.

#### 2.3.2. Drug-Likeness and ADME Studies 

Pharmacokinetic properties play a critical role in the discovery and development of the new drug candidates [48]. The successful transition of these new compounds to the development stage depends mainly on their pharmacokinetics and pharmacodynamics properties [49]. In this study, the molecular properties related to the pharmacokinetic and drug-likeness parameters of compounds **8a–i** were evaluated and compared with those of ibuprofen, ketoprofen, and compounds **3**. The study of the molecular properties was calculated using SwissADME [50] and Molsoft L.L.C. The results are presented in Table 5.

Based on Lipinski’s rule, an orally active drug has a total number of hydrogen bond donors ≤ 5, hydrogen bond acceptors ≤ 10, logP < 5, and molecular weight < 500 da [51]. An investigation of physicochemical parameters of **8a–i** revealed that their molecular weights are above 500 daltons. However, the molecular weights are less than that of compound **VI**. The polar surface areas of **8a–i** are either comparable or lower than that of compound **VI**. The calculated logP values of the nine hybrids are in the range of 2.12–3.73 compared to compound **VI** (logP = 3.71). The results also showed that compound **8a**,**c** have no violation from Lipinski’s rule, while all of the remaining compounds showed only one violation (MW > 500).

The tested hybrids also showed drug-likeness scores (DLS) in the range of 1.06−2.03 compared to 1.01 for compound **VI**. The fraction of **8a–i** that can undergo GIT absorption was calculated following the previous report [52]. The results revealed that 80.10–84.75% of **8a–i** could be absorbed from GIT compared to compound **VI** (80.40%).

The detailed results of the ADME study are provided in Appendix A.

## 3. Conclusions

The design of anticancer-NSAID hybrids could be used to avoid some of the problems encountered in combination therapy. Many of the hybrids combining ibuprofen with anticancer agents have displayed promising in vitro cytotoxic activities. In the current study, eight hybrids bearding the NSAIDs (ibuprofen and ketoprofen) with five pyrrolizine/indolizine derivatives (**5a–e**) were designed and synthesized. A structural elucidation of the new hybrids was confirmed using IR, mass, ^1^H-NMR, ^13^C NMR and DEPT C^135^ spectra. The new hybrids were evaluated for their antiproliferative activities against three (MCF-7, A549, and HT-29) cancer cell lines using a quick screening assay. The results revealed a 4–71% inhibition in the growth of the cancer cells, where MCF-7 cells were the most sensitive. Based on the results of this assay, compounds **8a**,**e**,**f**, the most active against MCF-7 cells were selected for additional biological evaluations. The cytotoxic activity of **8a**,**e**,**f** against MCF-7 cells revealed IC_50_ values in the range of 1.07–7.61. Cell cycle analysis of MCF-7 cells treated with the three hybrids at 5 μM revealed pro-apoptotic increase in cells at preG1 and cell cycle arrest at the G1 and S phases. In addition, the three hybrids induced early apoptotic events in MCF-7 cells. A molecular docking study of the three hybrids into the active sites of COX-1/2 was also performed. The results revealed higher binding free energies for the three hybrids toward COX-1/2 compared to the parent compounds **5a**,**c** and the co-crystallized ligands, ibuprofen and SC-558. The results also indicated the higher binding free energies of the three hybrids toward COX-2 over COX-1. An analysis of the binding modes of the best fitting conformation of the three hybrids into COX-2 revealed partial superposition with SC-558 and formation of hydrogen bonds, electrostatic, and hydrophobic interactions with the key amino acid such as His90, and Arg513. The new hybrids also showed drug-likeness score of 1.06–2.03. The above results support the future investigation of compounds **8a**,**e**,**f** as potential anticancer candidates.

## 4. Experimental Protocol

### 4.1. Chemistry

Chemical reagents and solvents were obtained from commercial sources. Solvents are dried by standard methods when necessary. The purity of the new compounds was checked with TLC. Melting points (m.p.) are uncorrected and were determined by IA 9100MK-Digital melting point apparatus (Cole-Parmer, East Bunker Ct Vernon Hills, IL, USA). Infrared spectra (IR) were recorded using BRUKER TENSOR 37 spectrophotometer (Bruker, Billerica, MA, USA). The proton magnetic spectra were recorded on BRUKER AVANCE III at 500 MHz, ^13^C NMR (125 MHz), and DEPT C^135^ (125 MHz). Mass spectra were recorded on Agilent UPLC/MS/MS 1260 infinity II with 6420 Triple quad LC/MS detector at Faculty of Pharmacy, Minia University, Minia, Egypt.

Compounds **2a**,**b** [34], and **4a–d** [35], and **5a–e** [36] were prepared according to the previous reports.

#### 4.1.1. General Procedure (A) for Preparation of Compounds (**8a–i**) 

The synthesis of the hybrids **8a–i** was achieved using the same reaction conditions applied in the synthesis of compound **8e** [28].

##### Preparation of Compounds **7a,b**

Thionyl chloride (1 gm, 8.41 mmol) was added to 5.9 mmol of (±)-ibuprofen/ketoprofen in a dry round-bottom flask. The reaction mixture was refluxed on a steam bath. After for 2 h reflux, the excess thionyl chloride was removed under reduced pressure. The residue (ibuprofen/ketoprofen acetyl chloride) was dissolved in methylene chloride (30 mL).

##### Preparation of Compounds **8a–j**

A solution of compound **5a–e** (3 mmol) in methylene chloride (30 mL) was added dropwise to the acid chloride solution obtained from the appropriate NSAIDs. To the reaction mixture, 0.5 mL of TEA was added, and the mixture was stirred for 2 h at rt. The reaction mixture was left to stand overnight. The solvent was evaporated under reduced pressure and the solid residue obtained was washed with 10 mL of aqueous sodium carbonate 2% and left to dry. The dry residue was recrystallized from ethanol-acetone.

*(R,S)-7-Cyano-6-(2-(4-isobutylphenyl)propanamido)-N-(4-methoxyphenyl)-2,3-dihydro-1H-pyrrolizine-5-carboxamide (**8a**)* The title compound was prepared from the reaction of compound **5a** (0.89 g, 3 mmol) with the acid chloride obtained from the reaction of thionyl chloride and ibuprofen (1.22 g, 5.9 mmol) according to the general procedure A. Compound **8a** was obtained as white solid product, m.p. 232–4 °C, yield 67%. IRυ_max_/cm^−1^ 3409, 3268 (NHs), 2952, 2868, 2832 (C-H aliphatic), 2221 (CN), 1659 (C=O), 1601, 1544, 1511 (C=C, C=N), 1425, 1308, 1248 (C-N, C-O). ^1^H-NMR (DMSO-*d_6_*, 500 MHz, *δ* ppm): *δ* 0.88 (d, 6H, *J* = 6.2 Hz, CH(CH_3_)_2_), 1.61 (d, 3H, *J* = 6.6 Hz, CHCH_3_), 1.77–1.85 (m, 1H, CH(CH_3_)_2_), 2.39–2.45 (m, 4H, pyrrolizine CH_2_-2 + Ph-CH_2_), 2.84 (t, 2H, *J* = 7.2 Hz, pyrrolizine CH_2_-1), 3.79 (s, 3H, OCH3), 4.16–4.29 (m, 3H, CHCH_3_+pyrrolizine CH_2_-3), 6.74 (d, 2H, *J* = 7.8 Hz, Ph (A) CH-3+CH-5), 7.04 (d, 2H, *J* = 7.02 Hz, Ph (B) CH-3+CH-5), 7.21 (d, 2H, *J* = 7.8 Hz, Ph (A) CH-2+CH-6), 7.26 (d, 2H, *J* = 7.1 Hz, Ph (B) CH-2+CH-6), 8.21 (s, 1H, CONHCH), 9.03 (s, 1H, CONHPh). ^13^C-NMR (DMSO, 125 MHz, *δ* ppm): *δ* 18.35, 22.39, 24.79, 25.45, 30.15, 45.03, 46.52, 49.41, 55.42, 84.44, 113.92, 114.11, 119.83, 121.29, 124.84, 127.38, 129.79, 130.82, 137.08, 141.13, 145.60, 146.13, 157.48, 176.88. DEPT C^135^ (DMSO, 125 MHz, *δ* ppm): *δ* 18.35, 22.39, 24.79, 25.46, 30.16, 45.03, 46.51, 49.41, 55.42, 113.91, 121.28, 127.38, 129.79. LC-MS: *m/z* 471.40 [M-13]^+^. Anal. Calcd. for C_29_H_32_N_4_O_3_ (484.59): C, 71.88; H, 6.66; N, 11.56. Found C, 71.37; H, 6.42; N, 10.70. 

*(R,S)-6-(2-(3-Benzoylphenyl)propanamido)-7-cyano-N-(4-methoxyphenyl)-2,3-dihydro-1H-pyrrolizine-5-carboxamide (**8b**)* The title compound was prepared from the reaction of compound **5b** (0.89 g, 3 mmol) with the acid chloride obtained from the reaction of thionyl chloride and ketoprofen (1.45 g, 5.9 mmol) according to the general procedure A. Compound **8b** was obtained as white solid product, m.p. 246–8 °C, yield 64%. IRυ_max_/cm^−1^ 3284, 3203 (NHs), 3085, 3003, 2968, 2841 (C-H aliphatic), 2211 (CN), 1698 (C=O), 1587, 1555, 1508 (C=C, C=N), 1429, 1302, 1230 (C-N, C-O). ^1^H-NMR (CDCl_3_, 500 MHz, *δ* ppm): *δ* 1.62 (d, 3H, *J* = 7.1 Hz, CHCH_3_), 2.38-2.43 (m, 2H, pyrrolizine CH_2_-2), 2.83 (t, 2H, *J* = 7.5 Hz, pyrrolizine CH_2_-1), 3.71 (s, 3H, OCH_3_), 3.99 (q, 1H, *J* = 6.7 Hz, CHCH_3_), 4.17–4.27 (m, 2H, pyrrolizine CH_2_-3), 6.69 (d, 2H, *J* = 8.3 Hz, aromatic Hs), 7.17 (d, 2H, *J* = 8.3 Hz, aromatic Hs), 7.31 (t, 1H, *J* = 7.5 Hz, aromatic H), 7.42 (t, 2H, *J* = 7.3 Hz, aromatic Hs), 7.55 (t, 2H, *J* = 8.1 Hz, aromatic Hs), 7.64 (d, 1H, *J* = 7.5 Hz, aromatic H), 7.68 (d, 2H, *J* = 7.5 Hz, aromatic Hs), 7.86 (s, 1H, Ph (B) CH-2), 8.94 (s, 1H, CONHCH), 9.02 (s, 1H, CONHPh). ^13^C-NMR (CDCl_3_, 125 MHz, *δ* ppm): *δ* 18.45, 24.79, 25.47, 46.49, 49.41, 55.40, 84.54, 113.96, 114.18, 119.73, 121.12, 124.90, 128.32, 128.76, 129.15, 129.55, 130.09, 130.76, 131.62, 132.64, 137.23, 137.94, 140.74, 145.72, 156.29, 157.47, 176.03, 196.78. DEPT C^135^ (CDCl_3_, 125 MHz, *δ* ppm): *δ* 18.46, 24.79, 25.47, 46.48, 49.41, 55.40, 113.96, 121.11, 128.33, 128.76, 129.15, 129.56, 130.10, 131.63, 132.65. LC-MS: *m/z* 531.40 [M-H]^+^. Anal. Calcd. for C_32_H_28_N_4_O_4_ (532.59): C, 72.16; H, 5.30; N, 10.52. Found C, 71.84; H, 4.98; N, 11.05. 

*(R,S)-7-Cyano-N-(4-fluorophenyl)-6-(2-(4-isobutylphenyl)propanamido)-2,3-dihydro-1H-pyrrolizine-5-carboxamide (**8c**)* The title compound was prepared from the reaction of compound **5b** (0.85 g, 3 mmol) with the acid chloride obtained from the reaction of thionyl chloride and ibuprofen (1.22 g, 5.9 mmol) according to the general procedure A. Compound **8c** was obtained as white solid product, m.p. 239–41 °C, yield 56%. IRυ_max_/cm^−1^ 3408, 3259 (NHs), 3065, 3046, 3013, 2949 (C-H aliphatic), 2223 (CN), 1662 (C=Os), 1612, 1548, 1510 (C=C, C=N), 1466, 1425, 1317, 1277 (C-N, C-O). ^1^H-NMR (DMSO-*d_6_*, 500 MHz, *δ* ppm): *δ* 0.80 (d, 6H, *J* = 5.6 Hz, CH(CH_3_)_2_), 1.42 (d, 3H, *J* = 6.0 Hz, CHCH_3_), 1.69–1.77 (m, 1H, CH(CH_3_)_2_), 2.35 (d, 2H, *J* = 6.7 Hz, Ph-CH_2_), 2.40–2.46 (m, 2H, pyrrolizine CH_2_-2), 2.97 (t, 2H, *J* = 6.6 Hz, pyrrolizine CH_2_-1), 3.89 (q, 1H, *J* = 7.0 Hz, CHCH_3_), 4.24 (t, 2H, *J* = 6.4 Hz, pyrrolizine CH_2_-3), 7.00 (d, 2H, *J* = 6.9 Hz, Ph (B) CH-3+CH-5), 7.11 (t, 2H, *J* = 7.9 Hz, Ph (A) CH-3+CH-5), 7.28 (d, 2H, *J* = 6.9 Hz, Ph (B) CH-2+CH-6), 7.38–7.41 (m, 2H, Ph (A) CH-2+CH-6), 9.43 (s, 1H, CONHCH), 10.20 (s, 1H, CONHPh). ^13^C-NMR (DMSO-*d_6_*, 125 MHz, *δ* ppm): *δ* 18.88, 22.60, 24.80, 25.66, 30.03, 44.67, 45.27, 49.70, 84.65, 114.92, 115.71 (d, *J* = 22.3 Hz, Ph (A) CH-3+CH-5), 118.65, 121.64 (d, *J* = 8.2 Hz, Ph (A) CH-2+CH-6), 126.78, 127.52, 129.40, 135.01 (d, *J* = 2.3 Hz, Ph (A) C-1), 138.63, 140.09, 146.43, 157.59, 158.69 (d, *J* = 240.2 Hz, Ph (A) C-4), 175.06. LC-MS: *m/z* 471.10 [M-H]^+^. Anal. Calcd. for C_28_H_29_FN_4_O_2_ (472.55): C, 71.17; H, 6.19; N, 11.86. Found C, 70.78; H, 5.81; N, 11.65.

*(R,S)-6-(2-(3-Benzoylphenyl)propanamido)-7-cyano-N-(4-fluorophenyl)-2,3-dihydro-1H-pyrrolizine-5-carboxamide (**8d**)* The title compound was prepared from the reaction of compound **5b** (0.85 g, 3 mmol) with the acid chloride obtained from the reaction of thionyl chloride and ketoprofen (1.45 g, 5.9 mmol) according to the general procedure A. Compound **8d** was obtained as white solid product, m.p. 240–2 °C, yield 59%. IRυ_max_/cm^−1^ 3355, 3210 (NHs), 3025, 2996, 2879 (C-H aliphatic), 2219 (CN), 1663, 1634 (C=Os), 1616, 1564, 1505 (C=C, C=N), 1442, 1322, 1208 (C-N, C-O). ^1^H-NMR (DMSO-*d_6_*, 500 MHz, *δ* ppm): *δ* 1.47 (d, 3H, *J* = 6.0 Hz, CHCH_3_), 2.41–2.46 (m, 2H, pyrrolizine CH_2_-2), 2.97 (t, 2H, *J* = 6.7 Hz, pyrrolizine CH_2_-1), 4.05 (q, 1H, *J* = 6.8 Hz, CHCH_3_), 4.24 (t, 2H, *J* = 6.7 Hz, pyrrolizine CH_2_-3), 7.06 (t, 2H, *J* = 8.0 Hz, aromatic Hs), 7.31–7.34 (m, 2H, aromatic Hs), 7.46 (t, 1H, *J* = 7.2 Hz, aromatic H), 7.51–7.56 (m, 3H, aromatic Hs), 7.64–7.70 (m, 4H, aromatic Hs), 7.71 (s, 1H, Ph (B) CH-2). 9.35 (s, 1H, CONHCH), 10.23 (s, 1H, CONHPh). ^13^C-NMR (DMSO-*d_6_*, 125 MHz, *δ* ppm): *δ* 18.95, 24.80, 25.66, 45.45, 49.69, 84.51, 114.84, 115.74 (d, *J* = 22.3 Hz, Ph (A) CH-3+CH-5), 118.54, 121.55 (d, *J* = 7.9 Hz, Ph (A) CH-2+CH-6), 126.67, 128.93, 128.99, 129.06, 129.20, 130.06, 132.13, 133.20, 134.98 (d, *J* = 2.3 Hz, Ph (A) C-1), 137.39, 137.55, 141.91, 146.44, 157.61, 158.65 (d, *J* = 240.6 Hz, Ph (A) C-4), 174.38, 196.09. DEPT C^135^ (DMSO-*d_6_*, 125 MHz, *δ* ppm): *δ* 18.95, 24.80, 25.66, 45.45, 49.69, 115.74 (d, *J* = 22.3 Hz, Ph (A) CH-3+CH-5), 121.55 (d, *J* = 7.9 Hz, Ph (A) CH-2+CH-6), 128.93, 128.99, 129.06, 129.20, 130.06, 132.13, 133.20. LC-MS: *m/z* 519.10 [M-H]^+^. Anal. Calcd. for C_31_H_25_FN_4_O_3_ (520.55): C, 71.53; H, 4.84; N, 10.76. Found C, 71.32; H, 4.46; N, 11.23.

*(R,S)-6-(2-(3-Benzoylphenyl)propanamido)-N-(4-bromophenyl)-7-cyano-2,3-dihydro-1H-pyrrolizine-5-carboxamide (**8f**)*The title compound was prepared from the reaction of compound **5c** (1.04 g, 3 mmol) with the acid chloride obtained from the reaction of thionyl chloride and ketoprofen (1.45 g, 5.9 mmol) according to the general procedure A. Compound **8f** was obtained as white solid product, m.p. 229–31 °C, yield 61%. IRυ_max_/cm^−1^ 3401, 3270 (NHs), 3063, 2965, 2874 (C-H aliphatic), 2222 (CN), 1659 (C=O), 1593, 1520 (C=C, C=N), 1423, 1315, 1270 (C-N, C-O). ^1^H-NMR (DMSO-*d_6_*, 500 MHz, *δ* ppm): *δ* 1.48 (d, 3H, *J* = 6.9 Hz, CHCH_3_), 2.44 (m, 2H, pyrrolizine CH_2_-2), 2.98 (t, 2H, *J* = 7.4 Hz, pyrrolizine CH_2_-1), 4.05 (q, 1H, *J* = 6.8 Hz, CHCH_3_), 4.25 (t, 2H, *J* = 7.1 Hz, pyrrolizine CH_2_-3), 7.29 (d, 2H, *J* = 8.5 Hz, aromatic Hs), 7.40 (d, 2H, *J* = 8.5 Hz, aromatic Hs), 7.47 (t, 1H, *J* = 7.6 Hz, aromatic H), 7.52–7.56 (m, 3H, aromatic Hs), 7.65–7.71 (m, 4H, aromatic Hs), 7.78 (s, 1H, Ph (B) CH-2). 9.43 (s, 1H, CONHCH), 10.24 (s, 1H, CONHPh). ^13^C-NMR (DMSO-*d_6_*, 125 MHz, *δ* ppm): *δ* 18.98, 24.82, 25.65, 45.46, 49.72, 84.57, 114.79, 115.71, 118.43, 121.60, 126.93, 128.96, 129.00, 129.05, 129.19, 130.04, 131.94, 132.15, 133.17, 137.42, 137.54, 138.04, 141.93, 146.59, 157.73. 174.36, 196.06. DEPT C^135^ (DMSO-*d_6_*, 125 MHz, *δ* ppm): *δ* 18.98, 24.82, 25.66, 45.46, 49.72, 121.60, 129.01, 129.19, 130.05, 131.94, 132.15, 133.17. LC-MS: *m/z* 581.20 [M+H]^+^. Anal. Calcd. for C_31_H_25_BrN_4_O_3_ (581.46): C, 64.03; H, 4.33; N, 9.64. Found C, 63.85; H, 4.69; N, 10.09.

*(R,S)-N-(4-Chlorophenyl)-1-cyano-2-(2-(4-isobutylphenyl)propanamido)-5,6,7,8-tetrahydroindolizine-3-carboxamide (**8g**)* The title compound was prepared from the reaction of compound **5d** (0.94 g, 3 mmol) with the acid chloride obtained from the reaction of thionyl chloride and ibuprofen (1.22 g, 5.9 mmol) according to the general procedure A. Compound **8c** was obtained as white solid product, m.p. 221–2 °C, yield 63%. IRυ_max_/cm^−1^ 3347, 3282 (NHs), 3059, 2952, 2907 (C-H aliphatic), 2223 (CN), 1655 (C=Os), 1603, 1571 (C=C, C=N), 1492, 1377 (C-N, C-O). ^1^H-NMR (DMSO-*d_6_*, 500 MHz, *δ* ppm): *δ* 0.81 (d, 6H, *J* = 4.7 Hz, CH(CH_3_)_2_), 1.39 (d, 3H, *J* = 7.0 Hz, CHCH_3_), 1.70–1.75 (m, 1H, CH(CH_3_)_2_), 1.78–1.81 (m, 2H, indolizine CH_2_-7), 1.87–1.90 (m, 2H, indolizine CH_2_-6), 2.34 (d, 2H, *J* = 7.0 Hz, Ph-CH_2_), 2.83 (t, 2H, *J* = 5.7 Hz, indolizine CH_2_-8), 3.85 (q, 1H, *J* = 7.2 Hz, CHCH_3_), 4.10 (t, 2H, *J* = 5.3 Hz, indolizine CH_2_-5), 6.95 (d, 2H, *J* = 7.9 Hz, Ph (B) CH-3+CH-5), 7.23 (d, 2H, *J* = 7.9 Hz, Ph (B) CH-2+CH-6), 7.31 (d, 2H, *J* = 8.7 Hz, Ph (A) CH-3+CH-5), 7.43 (d, 2H, *J* =8.7 Hz, Ph (A) CH-2+CH-6), 9.73 (s, 1H, CONHCH), 10.01 (s, 1H, CONHPh). ^13^C-NMR (DMSO, 125 MHz, *δ* ppm): *δ* 18.86, 18.94, 22.57, 22.60, 22.63, 30.01, 44.70, 45.20, 45.64, 88.57, 114.70, 121.29, 121.79, 124.00, 127.46, 127.68, 128.99, 129.30, 137.76, 138.53, 139.99, 140.55, 157.86, 175.04. DEPT C^135^ (DMSO, 125 MHz, *δ* ppm): *δ* 18.86, 18.94, 22.57, 22.60, 22.63, 30.01, 44.70, 45.20, 45.64, 121.29, 127.46, 128.99, 129.30. LC-MS: *m/z* 501.00 [M-H]^+^. Anal. Calcd. for C_29_H_31_ClN_4_O_2_ (503.04): C, 69.24; H, 6.21; Cl, 7.05; N, 11.14. Found C, 69.39; H, 5.76; N, 11.42.

*(R,S)-2-(2-(3-Benzoylphenyl)propanamido)-N-(4-chlorophenyl)-1-cyano-5,6,7,8-tetrahydroindolizine-3-carboxamide (**8h**)* The title compound was prepared from the reaction of compound **5d** (0.94 g, 3 mmol) with the acid chloride obtained from the reaction of thionyl chloride and ketoprofen (1.45 g, 5.9 mmol) according to the general procedure A. Compound **8h** was obtained as white solid product, m.p. 252–4 °C, yield 55%. IRυ_max_/cm^−1^ 3252, 3221 (NHs), 3084, 2974, 2959 (C-H aliphatic), 2217 (CN), 1662 (C=O), 1607, 1574, 1552 (C=C, C=N), 1489, 1314, 1259 (C-N, C-O). ^1^H-NMR (DMSO-*d_6_*, 500 MHz, *δ* ppm): *δ* 1.44 (d, 3H, *J* = 7.0 Hz, CHCH_3_), 1.76-1.81 (m, 2H, indolizine CH_2_-7), 1.86-1.91 (m, 2H, indolizine CH_2_-6), 2.82 (t, 2H, *J* = 6.0 Hz, indolizine CH_2_-8), 4.01 (q, 1H, *J* = 6.8 Hz, CHCH_3_), 4.10 (t, 2H, *J* = 5.5 Hz, indolizine CH_2_-5), 7.26 (d, 2H, *J* = 8.5 Hz, aromatic Hs), 7.38 (d, 2H, *J* = 8.5 Hz, aromatic Hs), 7.40–7.55 (m, 4H, aromatic Hs), 7.65–7.69 (m, 4H, aromatic Hs), 7.74 (s, 1H, Ph (B) CH-2), 9.70 (s, 1H, CONHCH), 10.11 (s, 1H, CONHPh). ^13^C-NMR (DMSO-*d_6_*, 125 MHz, *δ* ppm): *δ* 18.94, 22.56, 22.59, 45.36, 45.63, 52.92, 88.38, 114.65, 121.22, 123.94, 127.45, 128.85, 129.00, 129.09, 129.28, 130.05, 131.31, 132.09, 133.15, 137.45, 137.77, 139.35, 140.50, 141.85, 147.91, 164.53, 174.24, 196.05. DEPT C^135^ (DMSO-*d_6_*, 125 MHz, *δ* ppm): *δ* 18.93, 22.56, 22.59, 45.36, 45.63, 52.92, 121.22, 127.45, 128.85, 129.00, 129.10, 129.28, 130.06, 132.09, 133.15. LC-MS: *m/z* 549.20 [M-H]^+^. Anal. Calcd. for C_32_H_27_ClN_4_O_3_ (551.03): C, 69.75; H, 4.94; N, 10.17. Found C, 69.31; H, 4.67; N, 10.61. 

*(R,S)-N-(4-Bromophenyl)-1-cyano-2-(2-(4-isobutylphenyl)propanamido)-5,6,7,8-tetrahydroindolizine-3-carboxamide (**8i**)* The title compound was prepared from the reaction of compound **5e** (1.08 g, 3 mmol) with the acid chloride obtained from the reaction of thionyl chloride and ibuprofen (1.22 g, 5.9 mmol) according to the general procedure A. Compound **8i** was obtained as white solid product, m.p. 235–7 °C, yield 54%. IRυ_max_/cm^−1^ 3346, 3280 (NHs), 3056, 2952 (C-H aliphatic), 2222 (CN), 1656 (C=O), 1602, 1571, 1539 (C=C, C=N), 1489, 1310 (C-N, C-O). ^1^H-NMR (CDCl_3_, 500 MHz, *δ* ppm): *δ* 0.90 (d, 6H, *J* = 3.4 Hz, CH(CH_3_)_2_), 1.64 (d, 3H, *J* = 6.7 Hz, CHCH_3_), 1.80–1.88 (m, 3H, indolizine CH_2_-7+CH(CH_3_)_2_), 1.95–1.99 (m, 2H, indolizine CH_2_-6), 2.44 (d, 2H, *J* = 6.7 Hz, Ph-CH_2_), 2.86 (t, 2H, *J* = 6.1 Hz, indolizine CH_2_-8), 3.82 (q, 1H, *J* = 6.7 Hz, CHCH_3_), 4.14–4.27 (m, 2H, indolizine CH_2_-5), 7.05 (d, 2H, *J* = 7.0 Hz, Ph (B) CH-3+CH-5), 7.20 (d, 2H, *J* = 8.1 Hz, Ph (B) CH-2+CH-6), 7.28–7.31 (m, 2H, aromatic Hs), 7.38–7.41 (m, 3H, CONH+aromatic Hs), 9.67 (s, 1H, CONH). ^13^C-NMR (DMSO, 125 MHz, *δ* ppm): *δ* 18.09, 18.91, 22.38, 22.67, 22.73, 30.14, 44.97, 45.75, 46.81, 88.63, 113.75, 116.79, 121.01, 121.85, 123.45, 127.38, 129.97, 131.86, 136.40, 137.10, 140.49, 141.62, 157.55, 177.46. DEPT C^135^ (DMSO, 125 MHz, *δ* ppm): *δ* 18.09, 18.91, 22.38, 22.67, 22.74, 30.14, 44.97, 45.75, 46.81, 121.02, 127.38, 129.97, 131.86. LC-MS: m/z 547.00 [M+H]^+^. Anal. Calcd. for C_29_H_31_BrN_4_O_2_ (547.49): C, 63.62; H, 5.71; N, 10.23. Found C, 63.54; H, 5.46; N, 10.33.

### 4.2. Biological Evaluation 

#### 4.2.1. Antiproliferative Activity

##### Cell Culture

In this study, three cancer cell lines, MCF-7, A549, and HT-29 of the American Type Culture Collection (ATCC) were used in the evaluation of the antiproliferative activities of the new compounds. The cancer cells were cultured following the previously reported conditions [37].

##### Screening Assay

To evaluate the antiproliferative activities of the new compounds and their parent compounds, a quick screening MTT assay was used [37]. The cancer cell lines were treated with the hybrids **8a–i**, and the parent compounds, ibuprofen, ketoprofen and compounds **5a–e** at 5 μM. After 72 h of treatment, the absorbance of the purple formazan was determined using multi-plate reader at 570 nm and the growth% of the treated cells was determined and compared with the control.

##### MTT Assay

The antiproliferative activity of compounds **8a**,**e**,**f** was evaluated against MCF-7 cells using MTT assay [38]. First, 96-well plates were used for the seeding of the cancer cells for 24 h. The cancer cells were treated test compounds **8a**,**e**,**f**, and doxorubicin at different concentrations (0 µM–50 µM) for 72 h. The antiproliferative activity expressed as IC_50_ values for each of the four compounds were determined. 

#### 4.2.2. Cell Cycle Analysis

The effect of compounds **8a**,**e**,**f** on cell cycle perturbation of MCF-7 cells was evaluated using flowcytometry (BD FAC SCalibur flow cytometer, BD Biosciences, Franklin Lakes, NJ, USA). The cancer cell was cultured into 6 well plates, then were treated with each of the test compounds at 5 μM for 48 h. The assay was conducted following the previous report [36]. 

#### 4.2.3. Annexin V-FITC/PI Apoptosis Assay

Annexin V-FITC/PI double-staining assay was used to investigate the effect of compounds **8a**,**e**,**f** on apoptotic events in MCF-7 cells compared to the control. In 6 well plates, the cancer cells were seeded overnight. The cancer cells were treated with **8a**,**e**,**f** at 5 μM for 48 h. The assay was conducted according to our previous report [36]. NovoCyte flow cytometer (BD FAC SCalibur flow cytometer, BD Biosciences, Franklin Lakes, NJ, USA) was used to analyse the cancer cells with early/late apoptotic and necrotic changes. 

### 4.3. Computational Studies 

#### 4.3.1. Molecular Docking Studies

Compounds (**8a**,**e**,**f**) were evaluated in a comparative molecular docking study into COX-1 (pdb code: 1EQG) [43] and COX-2 (pdb code: 1CX2) [44]. In addition, the parent drugs of the three hybrids **5a,c** were also docked into the two COXs. The crystal structures of the two enzymes were obtained from protein data bank (http://www.rcsb.org/pdb) (accessed on 28 July 2021). AutoDock 4.2 [45] was used to dock the test compounds into the active sites of the two COXs. The study was performed following the previous report [46]. The generated 2/3D figures of the binding modes of the test compounds were created by discovery studio visualizer [47].

#### 4.3.2. Drug-Likeness and ADME Studies 

The molecular properties related to the pharmacokinetic/drug-likeness parameters of compounds **8a–i** were evaluated by SwissADME (http://www.swissadme.ch/) (accessed on 24 September 2021) [50] and Molsoft (http://molsoft.com/mprop/) (accessed on 7 August 2021) webservers in the calculation of the physicochemical properties. The compounds were either sketched directly or imputed as SMILES. The detailed results are provided in the Appendix A.

## Data Availability

The data supporting the reported results are avilable in the Appendix A.

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
