# Peer review of "Pyrrolizine/Indolizine-NSAID Hybrids: Design, Synthesis, Biological Evaluation, and Molecular Docking Studies"

_molecules, 2021, doi:10.3390/molecules26216582_

Round 1

Reviewer 1 Report

In the manuscript entitled "Pyrrolizine/indolizine-NSAID hybrids: design, synthesis, biological evaluation, and molecular docking studies", the authors present their work refining derivatives of common NSAIDs to create novel anti-cancer drugs. The work is comprehensive, and the manuscript will be quite impressive after a few minor revisions. Requested improvements for to this manuscript are quite minimal and fall into three main categories.

  1. Statistical analyses are incomplete for the data presented in Figures 4-6. The results are described as being, for example, “much higher” or “lower” than the controls. Are these differences statistically significant?  After completing the appropriate statistical analyses, the significantly different samples should be noted on the figures.  As a side note, potentially rearranging or strategically grouping the data (e.g., NSAID – parent compound – derivative compound, etc.) would aid not only in adding statistically significant findings, but also help the reader to make understand/remember the derivation of the experimental compounds. The included Tables may also benefit from selected statistical significance calculations.
  2. English editing is required. Incorrect vocabulary (e.g., “motility” on line 41), use of articles (i.e., “the”), and verb tenses and/or conjugations (e.g., “Several attempt was” on line 133) abound, especially in the Introduction section. Voice is inconsistent in places (e.g., compare amongst lines 126-146), and should be standardized. The text is overly informal in places and should be standardized to formal scientific writing (e.g., say “or” rather than using a / between words unless appropriate for the chemical description – e.g., see lines 104 and 121).
  3. Some minor omissions and/or typos are present. In particular:
    • Figures 1-3 are not properly described in their respective figure legends. Based on the main text, citations to previously published work may also be needed.
    • The discussion of the supplemental NMR figures is appropriately thorough. The only recommendation would be to specify the applicable supplementary figures in the discussion as they are summarized. The remaining supplementary figures should still be included in the note on lines 175-177. Further, the final sentence (lines 175-177) is not a complete paragraph, and can be moved to the end of the introductory sentences as part of that first paragraph (i.e., line 148).
    • Addition of more introductory detail in all sections would make the paper more understandable to a general audience. For example, a sentence or two of introductory text at the main subsections of section 2.2 would be useful. Also, in line 184, the three cell lines are introduced.  Unless the reader is familiar with those lines, they will have to dig all the way to the methods (line 648) to find the identity of those lines. Specifying the type of cancer represented by each cell line should come earlier in the text. Other examples are: description of SC-558 is not obvious or lacking – e.g., line 350 – and descriptions of “good” drug values – e.g., lines 422-433)
    • For Figure 8 and Table 8 – The Figure 8 panels do not have their labels A-D. Why does Table 3 use different compounds from Figure 8 (i.e., 8a, e, f vs. 8a, g, h)?
    • Are the docking studies theoretical structural overlays or crystallization studies? If they are theoretical structures, they should be labeled as such – including the applicable figures. The programs used to generate the structural images should be included in the Figure legends.
    • Please proofread for duplicated words – e.g., “cancer cancers” on line 50 and “results of the docking results” on line 347.

Author Response

Review 1

Comments and Suggestions for Authors

In the manuscript entitled "Pyrrolizine/indolizine-NSAID hybrids: design, synthesis, biological evaluation, and molecular docking studies", the authors present their work refining derivatives of common NSAIDs to create novel anti-cancer drugs. The work is comprehensive, and the manuscript will be quite impressive after a few minor revisions. Requested improvements for to this manuscript are quite minimal and fall into three main categories.

We highly appreciate the valuable comments of reviewer 1 and his valuable corrections that have been emerged after his careful and precise revision which would help in improving the quality of the manuscript. We indicated the revisions/corrections by a yellow highlighter in the revised manuscript. Below, are our responses to the comments, point-by-point.

Comment:

  1. Statistical analyses are incomplete for the data presented in Figures 4-6. The results are described as being, for example, “much higher” or “lower” than the controls. Are these differences statistically significant?  After completing the appropriate statistical analyses, the significantly different samples should be noted on the figures.  As a side note, potentially rearranging or strategically grouping the data (e.g., NSAID – parent compound – derivative compound, etc.) would aid not only in adding statistically significant findings but also help the reader to make understand/remember the derivation of the experimental compounds. The included Tables may also benefit from selected statistical significance calculations.

Response: We thank the reviewer for his comments. Statistical analysis was performed as requested and description was added in the figure legend. In addition, data of the tested compounds was rearranged as suggested

Comment:

  1. English editing is required. Incorrect vocabulary (e.g., “motility” on line 41), use of articles (i.e., “the”), and verb tenses and/or conjugations (e.g., “Several attempt was” on line 133) abound, especially in the Introduction section. Voice is inconsistent in places (e.g., compare amongst lines 126-146), and should be standardized. The text is overly informal in places and should be standardized to formal scientific writing (e.g., say “or” rather than using a / between words unless appropriate for the chemical description – e.g., see lines 104 and 121).

Response:

  • The indicated typo/grammar mistakes were corrected in the revised manuscript. In addition, the manuscript was also revised for any other typo/grammar mistakes before the re-submission.
  • The text on lines 126-146 was revised and corrected in the revised manuscript
  • The syntax of the sentence on line 133 was revised and corrected
  • The sentences on lines 104-107 were rewritten in formal scientific words
  • The sentences on lines 121-125 were rewritten in formal scientific words

Comment:

  1. Some minor omissions and/or typos are present. In particular:

Figures 1-3 are not properly described in their respective figure legends. Based on the main text, citations to previously published work may also be needed.

Response: The legends of figures 1-3 were updated with short description with citation of the previous work.

Comment

The discussion of the supplemental NMR figures is appropriately thorough. The only recommendation would be to specify the applicable supplementary figures in the discussion as they are summarized. The remaining supplementary figures should still be included in the note on lines 175-177. Further, the final sentence (lines 175-177) is not a complete paragraph and can be moved to the end of the introductory sentences as part of that first paragraph (i.e., line 148).

Response:

The number of the spectral figures in the supplementary data were added to the revised manuscript.

The final sentence (lines 175-177) was revised and corrected in the updated manuscript.

Comment

Addition of more introductory detail in all sections would make the paper more understandable to a general audience. For example, a sentence or two of introductory text at the main subsections of section 2.2 would be useful. Also, in line 184, the three cell lines are introduced.  Unless the reader is familiar with those lines, they will have to dig all the way to the methods (line 648) to find the identity of those lines. Specifying the type of cancer represented by each cell line should come earlier in the text. Other examples are: description of SC-558 is not obvious or lacking – e.g., line 350 – and descriptions of “good” drug values – e.g., lines 422-433)

Response:

An introductory sentence was added to section 2.2.

The identity/type of the three cancer cell lines were included in section 2.2.

SC-558 was defined in the text and in the legend of Table 4.

The criteria of the Lipinski's rule was briefly discussed in the revised manuscript.  

Comment

For Figure 8 and Table 8 – The Figure 8 panels do not have their labels A-D. Why does Table 3 use different compounds from Figure 8 (i.e., 8a, e, f vs. 8a, g, h)?

Response:

Figure 8 was revised and corrected

Table 3 was revised and corrected

Comment

Are the docking studies theoretical structural overlays or crystallization studies? If they are theoretical structures, they should be labeled as such – including the applicable figures. The programs used to generate the structural images should be included in the Figure legends.

Response:

The docking study is a theoretical study. The legend of Figs. 9-14 were updated to indicate that they were generated based on a theoretical docking study.

The program used to generate the docking figures was included in the legend o all the docking figure and in the text in section 2.3.1.

Comment

Please proofread for duplicated words – e.g., “cancer cancers” on line 50 and “results of the docking results” on line 347.

Response: The duplicated words on lines 50 and 347 were corrected.

Reviewer 2 Report

The manuscript „Pyrrolizine/indolizine-NSAID hybrids: design, synthesis, biological evaluation, and molecular docking studies“ by M. A. S. Abourehab and co-workers reports on the chemical synthesis, analytical characterization and biological investigation of pyrrolizine/indolizine hybrids with ibuprofen/ketoprofen derivatives. In an initial screening using MCF-7, A549 and HT-29 cells, the compounds 8a, 8e and 8f revealed to be most cytotoxic against MCF-7 cells. Therefore, these compounds (they fulfill the Lipinski rule as calculated) and this cell line was used for studying the mechanism of action. The investigations covered antiproliferative IC50 values, cell cycle analysis and the impact of apoptosis, while the compounds were active in the single-digit µM range and mainly exerted early apoptosis. Docking studies considering COX1 and COX2 complemented the experiments. Overall, 8a, 8e and 8f seem to be promising candidates for further research.

@ Introduction: There are several NSAIDs with acetylsalicylic acid being the most prominent representative. Much research was performed in recent years to exploit the effects of derivatives of ASA in order to treat cancer cell lines. The authors are kindly asked to consider these research in their introduction: J Med Chem, 2016, 59, 5, 1946-1959 Selenium NSAIDs; Int J Mol Sci, 2018, 19, 6, 1612 Platinum complexes; Dalton Trans, 2019, 48, 42, 15856-15868 Cobalt complexes; Molecules, 2020, 25, 17, 3849 Resveratrol hybrids. However, the authors should provide a reason, why they decided to study on ibuprofen/ketoprofen derivatives, although ASA derivatives showed very promising effects.

@ Figure 2: It is not clear, what is compound V and what is compound IV. Please clarify this.

@ Scheme 1: Why did the authors not synthesize all possible derivatives, e.g. n=1, Cl, 4-Isobutyl/3-Benzoyl? E.g. n=2, Br, 3-Bezoyl? n=2 OCH3, F, respectively, each 4-Isobutyl/3-Benzoyl? Omitting these derivatives limits the structure-activity-study.

Is it maybe possible to combine Figure 4, Figure 5 and Figure 6 in one single figure? This would allow for proper comparison between the different cell lines.

@ Table 1: Please provide a reference, indicating that the value of doxorubicin is in agreement with literature.

The authors discuss the different findings that COX may contribute to the anticancer effect or not.  It would be interesting to know the COX content of the used cell lines. HT-29 are considered to express high levels of enzymes, while MCF-7 are considered COX-negative (Dalton Trans, 2018, 47, 4341-4351).

The authors consider COX1 and/or COX2 as target of the compounds. Therefore, docking studies were performed. Actually, the potential of the compounds to inhibit the isoenzymes should be investigated using the isolated enzyme. The authors are kindly asked to check, if the enzyme is commercially available in-between. Due to the pandemic situation, it was not possible to order the enzyme some few weeks ago. If possible, the authors are asked to study at least 8a, 8e and 8f regarding the potency to inhibit the isoenzymes. It would be of interest, if there is a selectivity for COX2 since inhibition of COX1 causes limiting side effects of the gastrointestinal tract. The docking studies already indicate a stronger binding to COX2. The authors mention side effects of hybrid drugs in their introduction. However, maybe they can correlate to the COX expression of the cell lines.

Although there will be a close editing by the MDPI publisher, the authors are kindly asked to improve the quality of the manuscript considering a proper formatting. Optimization of English in same paragraphs is also recommended.

Author Response

The manuscript „Pyrrolizine/indolizine-NSAID hybrids: design, synthesis, biological evaluation, and molecular docking studies“ by M. A. S. Abourehab and co-workers reports on the chemical synthesis, analytical characterization and biological investigation of pyrrolizine/indolizine hybrids with ibuprofen/ketoprofen derivatives. In an initial screening using MCF-7, A549 and HT-29 cells, the compounds 8a, 8e and 8f revealed to be most cytotoxic against MCF-7 cells. Therefore, these compounds (they fulfill the Lipinski rule as calculated) and this cell line was used for studying the mechanism of action. The investigations covered antiproliferative IC50 values, cell cycle analysis, and the impact of apoptosis, while the compounds were active in the single-digit µM range and mainly exerted early apoptosis. Docking studies considering COX1 and COX2 complemented the experiments. Overall, 8a, 8e, and 8f seem to be promising candidates for further research.

Response: We highly appreciate the valuable comments of reviewer 2 and his valuable corrections that have been emerged after his careful and precise revision which would help in improving the quality of the manuscript. We indicated all the revisions/corrections by a yellow highlighter in the revised manuscript. Below, are our responses to the comments, point-by-point.

Comment

Introduction: There are several NSAIDs with acetylsalicylic acid being the most prominent representative. Much research was performed in recent years to exploit the effects of derivatives of ASA in order to treat cancer cell lines. The authors are kindly asked to consider these research in their introduction: J Med Chem, 2016, 59, 5, 1946-1959 Selenium NSAIDs; Int J Mol Sci, 2018, 19, 6, 1612 Platinum complexes; Dalton Trans, 2019, 48, 42, 15856-15868 Cobalt complexes; Molecules, 2020, 25, 17, 3849 Resveratrol hybrids. However, the authors should provide a reason, why they decided to study ibuprofen/ketoprofen derivatives, although ASA derivatives showed very promising effects.

Response: the first 3 articles were cited in the introduction in Fig. 1 and were described in the text. The reasons behind the use of ketoprofen and ibuprofen were discussed in the rational design

Comment

Figure 2: It is not clear, what is compound V and what is compound IV. Please clarify this.

Response: 

We thank the reviewer for this correction. Fig. V was revised and corrected

Comment

Scheme 1: Why did the authors not synthesize all possible derivatives, e.g. n=1, Cl, 4-Isobutyl/3-Benzoyl? E.g. n=2, Br, 3-Benzoyl? n=2 OCH3, F, respectively, each 4-Isobutyl/3-Benzoyl? Omitting these derivatives limits the structure-activity study.

Response: We tried to prepare the OCH3/F indolizine derivative (n = 2). However, we got uncyclized products. Although the reflux time was increased to 48 h. We have discussed in results in the manuscript “lines 148-153”. The initial plan also included the preparation of Br indolizine derivative (n = 2) with ketoprofen. We also found some difficulties when we tried to import δ-Valerolactam/bromoaniline. However, this derivative (Br indolizine-ketoprofen) could be investigated in a prespective study.  

Comment

Is it may be possible to combine Figure 4, Figure 5, and Figure 6 in one single figure? This would allow for proper comparison between the different cell lines.

Response: We thank the reviewer for this suggestion. However, when we tried to combine the three figures (4-6) into one figure we got a complicated figure with a lower resolution, that may be difficult for the reader to follow. Especially after the addition of the asterisks of the statistical analysis. So, we kept them as they are after updating them.

Comment

Table 1: Please provide a reference, indicating that the value of doxorubicin is in agreement with the literature.

Response: A  reference indicating the IC50 value of doxorubicin against MCF-7 cells was added

Comment

The authors discuss the different findings that COX may contribute to the anticancer effect or not.  It would be interesting to know the COX content of the used cell lines. HT-29 are considered to express high levels of enzymes, while MCF-7 are considered COX-negative (Dalton Trans, 2018, 47, 4341-4351).

Response: We agree with the reviewer that MCF-7 lacks COX-2 protein, while COX-1 is overexpressed in these cells. However, many of the COX-2 selective inhibitors were reported with anticancer activity against MCF-7 cells. In addition, some of these inhibitors exhibited higher cytotoxic activity against MCF7 cells than the other COX-2 -overexpressing breast cancer cell lines. Some references below support these findings.

https://www.ncbi.nlm.nih.gov/pmc/articles/PMC3086307/

https://www.oncotarget.com/article/23250/text/

https://pubmed.ncbi.nlm.nih.gov/21140284/

In addition, the results in our study suggested that compounds 8a,e,f deserve future studies as potential antiproliferative agents. This study would allow us to investigate the exact molecular target involved in the mechanism of action of these hybrids.

Comment

The authors consider COX1 and/or COX2 as targets of the compounds. Therefore, docking studies were performed. Actually, the potential of the compounds to inhibit the isoenzymes should be investigated using the isolated enzyme. The authors are kindly asked to check if the enzyme is commercially available in-between. Due to the pandemic situation, it was not possible to order the enzyme some few weeks ago. If possible, the authors are asked to study at least 8a, 8e and 8f regarding the potency to inhibit the isoenzymes. It would be of interest, if there is a selectivity for COX2 since inhibition of COX1 causes limiting side effects of the gastrointestinal tract. The docking studies already indicate a stronger binding to COX2. The authors mention side effects of hybrid drugs in their introduction. However, maybe they can correlate to the COX expression of the cell lines.

Response: We would like to thank reviewer 2 for this comment. Actually, we have planned to evaluate the inhibitory activity of the new hybrids against COX-1/2. However, due to the COVID19 pandemic, several difficulties emerged regarding the transfer of chemical, reagents, and COXs kits which need -80 C storage and transfer.  

Comment

Although there will be close editing by the MDPI publisher, the authors are kindly asked to improve the quality of the manuscript considering proper formatting. Optimization of English in the same paragraphs is also recommended.

Response: The manuscript was revised for any typo/grammar mistakes

Round 2

Reviewer 2 Report

The authors did a very diligent job to address the concerns mentioned before and they successfully improved the quality of the manuscript.

The optimization covered the embedding of recently published work on NSAID-derivatives developed for the treatment of several cancer cell lines. Figure 2 was also optimized and the suggestion to merge Figures 4-6 was rejected by the authors. They are totally right, as this would probably lead to an “overloaded” Figure. I apologize for my former suggestion and totally agree with the authors. Moreover, the authors clearly explained why several derivatives were not included in the current study. As suggested, a reference of the IC50 value of doxorubicin was added. Finally, the inhibition of COX isoenzymes was suggested. As supposed, delivery of isolated enzymes seems still not possible for months. However, the authors wish to consider that in their upcoming study, which focuses on a more detailed mechanistic investigation of compounds 8a, 8e, 8f.

As the authors may receive a proof after acceptation of their manuscript, they are kindly asked to have a close look at some typos/formal inconsistencies, e.g.,

# caption figure 1: …conjugates with anticancer activities… (delete “of”); IIa-d, Zeise’s Salt derivatives (not “IIa-c”);

# caption figure 2 VIIIb,c (c also bold);

# line 223: 13-48%; line 247: 47-49%; compare line 454: 80.10-84.75%

# caption figure 7 / caption Table 3: “(n = 3). Experiment was repeated 3x.” Isn’t this the same, or did you perform three experiments with three replicates each?

# Table 4, * different fonts.

In conclusion, I am pleased to suggest the manuscript for publication. All the best!